# Numerical Investigation of Interlaminar Stress of CRTS II Slab Ballastless Track Induced by Creep and Shrinkage of Concrete

**DOI:** 10.3390/ma15072480

**Published:** 2022-03-27

**Authors:** Zhihui Zheng, Peng Liu, Zhiwu Yu, Yachuan Kuang, Lei Liu, Sasa He, Xiaoqiang Zhang, Qianghui Li, Wen Xu, Maofeng Lv

**Affiliations:** 1School of Civil Engineering, Central South University, 22 Shaoshan Road, Changsha 410075, China; zzh174801009@csu.edu.cn (Z.Z.); liupeng868@csu.edu.cn (P.L.); 2015038@csu.edu.cn (Z.Y.); 216056@csu.edu.cn (L.L.); 2National Engineering Laboratory for High Speed Railway Construction, Central South University, Changsha 410075, China; 3Chenzhou Changxin Residence Technology Co., Ltd., 8 Huihuang Road, Chenzhou 423000, China; 4School of Physics and Technology, YiLi Normal University, 448 Jiefang Road, Yining 835000, China; 5Hunan Zhongda Design Institue Co., Ltd., 68 Shaoshan Road, Changsha 410075, China; hnzdsjy@yeah.net (S.H.); szwang863@163.com (Q.L.); 6Jinan Municipal Engineering Design & Research Institute Group Co., Ltd., Jinan 250004, China; zhszxq@jnszy.com; 7Railway Group 5 Mechanization of Engineering Co., Ltd., 32 HongtangCong Road, Hengyang 421002, China; miaolx863@163.com (W.X.); l15116277646@163.com (M.L.)

**Keywords:** ballastless track, shrinkage, creep, interfacial stress

## Abstract

The secondary development of ABAQUS was carried out to calculate the shrinkage and creep of concrete, and a finite element model of the China Railway Track System (CRTS) II slab ballastless track was established. Then, the interlaminar stress of CRTS II slab ballastless track at different ages of the track slab during laying (AOTSL) caused by concrete shrinkage and creep was studied. The obtained results showed that the stress redistribution occurred in the sliding layer, which resulted in the generation of a gap. Although the gap length was slightly reduced due to the shear cogging, the sliding layer at the slab edge is more prone to produce gaps. Under the effect of shrinkage and creep of the ballastless track, large additional shear stress, up to 0.676 MPa, was induced at the interface between CA mortar and the track slab. Meanwhile, the appearance of additional vertical and lateral forces of the shear cogging was caused by the shrinkage and creep of the ballastless track. Additionally, by further analysis, the recommended AOTSL ranges from 120 days to 180 days.

## 1. Introduction

In order to speed up economic development and facilitate people’s travel, the high-speed railway was vigorously developed during the preceding decades. Generally, to eliminate the pulverization of ballast, the slab ballastless track is widely used in the high-speed railway, including the China Railway Track System (CRTS) I, CRTS II, and CRTS III slab ballastless track in China. However, a large number of diseases were observed in the operation stage, such as the surface crack of the track slab, the crack of the base plate, the interface separation between the filling layer and the track slab, etc. [1,2,3,4]. Among them, the interface separation of the CRTS II slab ballastless track is the most significant [5], which seriously affects the dynamic characteristics of the slab track and causes potential damage [6]. Specifically, the interface separation may further result in the emergence of the void in the filling layer (Figure 1), which seriously affects the dynamic characteristics of the train–track–bridge coupled system. The existing research results showed that temperature is a key parameter to affect the ballastless track [7,8,9], so the effect of temperature on the interface stress has attracted the attention of many researchers.

To date, much effort has been directed towards the exploration of interface separation related to the effect of the temperature gradient [10,11,12,13,14,15]. For example, Refs. [10,11] established a model of the three-dimensional transient heat transfer of the ballastless track based on the measured temperature data to reveal the evolution process of interface damage under a daily temperature. The results showed that the damage separation might occur at the interface on both sides of the track slab under a daily temperature gradient. Further, a reasonable track laying time was obtained [12]. Cai et al. [13] analyzed the arching mechanism for the joints of the CRTS II ballastless track slab under temperature load by establishing a finite element model. It showed that damage occurs in the joints when the temperature of the track slab exceeds 37 °C. In addition, it was also found that the interface separation near the joints and the track deformation emerges due to the temperature gradient and pier settlement, respectively. Li et al. [14] performed an analysis of the interface damage between cement asphalt mortar (CA mortar) and the track slab of CRTS II ballastless track under the temperature gradient by using different interface strengths. Additionally, Zhang et al. [15] considered the viscoelastic properties of CA mortar and systematically studied the initial mechanism of interlayer debonding. The results showed that the use of the viscoelastic model leads to a significant increase in the height and width of the interlayer gap, but the temperature gradient of the initial debonding keeps unchanged. The above research manifested that the deformation of each layer of the ballastless track is uncoordinated under the effect of the temperature gradient, which leads to the interface of additional stress and the emergence of interface damage.

However, the interface damage of the CRTS II slab ballastless track is not only related to the temperature effect but also to its structural form and the differences in the age and property of materials. CRTS II slab ballastless track is mainly composed of the track slab, CA mortar layer, and the base plate. Among them, the base plate is connected with the bridge (mainly prefabricated simply supported girder) through the shear cogging and the sliding layer, and the base plate and the track slab are connected by CA mortar. Therefore, there are differences in the constraints of each layer of the ballastless track. In addition, the track slab is a prefabricated prestressed component, while the base slab and CA mortar layer are cast-in-place components, so the ballastless track is essentially a new–old concrete structure. For this type of multi-layer new–old concrete structure, each layer inevitably produces inconsistent material shrinkage and creep strains, which in turn generates additional stress at the interface of the new–old concrete [16,17,18]. Research [19,20] focused on the stress at the joints of the new–old beam concrete under the difference of shrinkage and creep. It showed that significant tensile stresses were generated both at joints and new beams. Further, the interface crack of the new–old concrete (Figure 2) emerges due to the excessive differences of shrinkage and creep [21,22,23,24]. Therefore, it can be seen that the difference in shrinkage and creep of the new–old concrete has a significant adverse effect on the engineering structure. However, CRTS II slab ballastless track is also as a new–old concrete structure, whereas that for present researches, they are in deficiency of elaborating this influence such as interface disease and the additional force of the connector (i.e., shear cogging), resulting in the difference in shrinkage and creep of the new–old concrete. Therefore, the interlaminar stress of the CRTS II slab ballastless track should be studied in depth under the effect of shrinkage and creep of the ballastless track.

In this study, the secondary development of the commercial finite element software ABAQUS (2020 version, 2020, Providence, RI, USA) was realized to calculate the shrinkage and creep of concrete. Then, a finite element model of CRTS II slab ballastless track was established to analyze interlaminar stresses, which include the stress of the sliding layer, the interface stress between CA mortar and the track slab, and the additional force of the shear cogging under the effect of shrinkage and creep of the ballastless track. Furthermore, the influence of the shear cogging on the gap length of the sliding layer and the interface stress were analyzed. Lastly, based on the results of interlaminar stress, the reasonable age of the track slab during laying (AOTSL) was obtained.

## 2. Calculation of Concrete Shrinkage and Creep in ABAQUS

### 2.1. Secondary Development of ABAQUS

To date, the analysis method of shrinkage and creep effect is divided into the theoretical analysis method and the finite element method. Because the mechanism of concrete shrinkage and creep has not been fully understood, the existing methods still have difficulties in accurately predicting the shrinkage and creep effects of the structure [25]. However, for complex structures, the finite element method is still the primary method. ABAQUS was widely used in civil engineering with its superior nonlinear analysis capabilities. Although there are many calculation modules in ABAQUS, it does not include a module to calculate shrinkage and creep of concrete. Therefore, the calculation of concrete shrinkage and creep still requires secondary development.

Thus far, the analysis of concrete shrinkage and creep by ABAQUS is based on the concrete stress–strain expression in integral form, which is then divided into several periods and converted into an algebraic form for solving [26,27,28,29]. When the finite element method is used to analyze the creep of structures, the Dirichlet series is usually used to express the creep coefficient to avoid storing the entire time history [30]. Especially, the creep coefficient expressed by the Dirichlet series can accurately reflect the difference in creep effect at different loading ages [26].

According to the principle of linear superposition, the total strain caused by concrete shrinkage and creep is expressed as follows [30]:(1)$$\varepsilon \left(t\right)=\sigma \left(\tau_{0}\right)c\left(t,\tau_{0}\right)+\int_{\tau_{0}}^{t}\left[\frac{1}{E\left(\tau \right)}+c\left(t,\tau \right)\right]d\sigma \left(\tau \right)+\varepsilon_{sh}\left(t\right)$$
(2)c(t,τ)=φ(t,τ)/E(τ)
where $\sigma \left(\tau_{0}\right)$ is the concrete stress caused by loading at time $\tau_{0}$; E(τ) is the elastic modulus of the concrete at time τ; $\varepsilon_{sh}\left(t\right)$ is the shrinkage strain of the concrete at time *t*; φ(t,τ) is the creep coefficient of the concrete.

The creep coefficient is converted into:(3)$$\varphi \left(t,\tau \right)=\sum_{j=1}^{m}a_{j}\left(\tau \right)\left(1-e^{-\lambda_{j}\left(t-\tau \right)}\right)$$
where $a_{j}\left(\tau \right)$ is the fitting parameter; $\lambda_{j}$ is a constant depending on the creep coefficient.

According to Equation (1), the algebraic expression of concrete shrinkage and creep strain increment can be expressed as follows [31]:(4)$$\begin{array}{l} \varepsilon_{n}=\sum_{i=0}^{n-1}\sigma_{i}\left[c\left(t_{n},t_{i}\right)-c\left(t_{n-1},t_{i}\right)\right]+\left[\varepsilon_{sh}\left(t_{n}\right)-\varepsilon_{sh}\left(t_{n-1}\right)\right]+\sum_{i=1}^{n-1}\int_{t_{i-1}}^{t_{i}}\left[c\left(t_{n},\tau \right)-c\left(t_{n-1},\tau \right)\right]d\sigma^{*}\left(\tau \right)\\ \;\;\;\;\;\;+\int_{t_{n-1}}^{t_{n}}\left[\frac{1}{E\left(\tau \right)}+c\left(t_{n},\tau \right)\right]d\sigma^{*}\left(\tau \right)\\ \;\;\;\;=\sum_{i=0}^{n-1}\sigma_{i}\left[c\left(t_{n},t_{i}\right)-c\left(t_{n-1},t_{i}\right)\right]+\left[\varepsilon_{sh}\left(t_{n}\right)-\varepsilon_{sh}\left(t_{n-1}\right)\right]+\sum_{i=1}^{n-1}\sigma_{i}^{*}\left[\frac{1}{E_{\varphi }\left(t_{n},t_{i-1}\right)}-\frac{1}{E_{\varphi }\left(t_{n-1},t_{i-1}\right)}\right]\\ \;\;\;\;\;\;+\frac{\sigma_{n}^{*}}{E_{\varphi }\left(t_{n},t_{n-1}\right)} \end{array}$$
(5)$$E_{\varphi }\left(t_{n-1},t_{i-1}\right)=\frac{E\left(t_{n-1}\right)}{1+\chi \varphi \left(t_{n-1}\right)}$$
where $\sigma_{i}$ is the concrete stress caused by the external load at time *t_i_*; $\sigma_{i}^{*}$ is the creep stress increment of concrete from time *t_i_*_−1_ to *t_i_*; χ is the aging coefficient, recommended value of 0.82 [32].

By substituting Equation (3) into Equation (4), the corresponding formulation can be obtained
(6)$$\begin{array}{l} \varepsilon_{n}=\sum_{i=0}^{n-1}\frac{\sigma_{i}}{E\left(t_{i}\right)}\sum_{j=1}^{m}a_{j}\left(t_{i}\right)e^{-\lambda_{j}\left(t_{n-1}-t_{i}\right)}\left(1-e^{-\lambda_{j}\Delta t_{n}}\right)+\left[\varepsilon_{sh}\left(t_{n}\right)-\varepsilon_{sh}\left(t_{n-1}\right)\right]\\ \;\;\;\;\;\;+\sum_{i=0}^{n-1}\chi \frac{\sigma_{i}^{*}}{E\left(t_{i}\right)}\sum_{j=1}^{m}a_{j}\left(t_{i-1}\right)e^{-\lambda_{j}\left(t_{n-1}-t_{i-1}\right)}\left(1-e^{-\lambda_{j}\Delta t_{n}}\right)+\frac{\sigma_{n}^{*}}{E_{\varphi }\left(t_{n},t_{n-1}\right)} \end{array}$$

Equation (6) is the creep stress increment to be solved, which is transformed into the following recursive form.
(7)$$\sigma_{n}^{*}=E_{\varphi }\left(t_{n},t_{n-1}\right)\left(\varepsilon_{n}-\Delta \varepsilon_{n}\right)$$
(8)$$\Delta \varepsilon_{n}=\varepsilon_{sh}\left(t_{n}\right)-\varepsilon_{sh}\left(t_{n-1}\right)+\sum_{j=1}^{m}A_{j,n}\left(1-e^{-\lambda_{j}\Delta t_{n}}\right)$$
(9)$$A_{j,n}=A_{j,n-1}e^{-\lambda_{j}\Delta t_{n-1}}+\sigma_{n-1}a_{j}\left(t_{n-1}\right)/E\left(t_{n-1}\right)+\chi \sigma_{n-1}^{*}a_{j}\left(t_{n-2}\right)e^{-\lambda_{j}\Delta t_{n-2}}/E\left(t_{n-2}\right)$$

The calculation of concrete shrinkage and creep strain can be realized by Equations (7)–(9). The USDFLD and UNEXPAN subroutines in ABAQUS are used to compile the creep and shrinkage computation modules by Fortran. The subroutine USDFLD is to define the time field variables to realize the time-varying elastic modulus. The subroutine UNEXPAN is called to store $\Delta \varepsilon_{n}$. Thus, the shrinkage and creep effects of the structure can be analyzed.

### 2.2. Shrinkage and Creep Models of Concrete

Shrinkage and creep models are the basis for analyzing structural shrinkage and creep, but there are variances between different models. Therefore, it is important to choose an appropriate model. Currently, creep models widely used in the world include the CEB-FIP MC 90 model, the Muller model, the ACI 209R-92 model, the B3 model, the B4 model, and the GL2000 model [33]. Among them, the B3 model and the GL2000 model have the best prediction accuracy, followed by the CEB-FIP MC 90 model [33]. The CEB-FIP MC 90 model has good prediction accuracy in the natural environment of China [34]. However, the CEB-FIP MC 90 model is only suitable for the concrete below C50. In addition, Ref. [35] showed that the GL2000 model is recommended for creep prediction of high-strength concrete. Therefore, the GL2000 model is used to calculate the shrinkage strain and the creep coefficient of concrete.

The GL2000 model adopts the following expressions for shrinkage [36]:(10)$$\varepsilon_{sh}\left(t,t_{c}\right)=\varepsilon_{\mathrm{shu}}\left(1-1.18RH^{4}\right)\sqrt{\left[\frac{t-t_{c}}{t-t_{c}+0.15{\left(V/S\right)}^{2}}\right]}$$
(11)$$\varepsilon_{\mathrm{shu}}=1000K\sqrt{\frac{30}{f_{\mathrm{cm}28}}}\times 10^{-6}$$
(12)$$f_{\mathrm{cm}28}=1.1f_{\mathrm{ck}28}+5$$
where *t* is the calculating time of concrete age, day (d); *t_c_* is the concrete age at the end of moist curing, the recommended value of 3 d; *RH* is the relative humidity of the ambient environment; *V/S* is the volume–surface ratio of the component; *K* is the coefficient related to the type of cement, 1.0 for class I cement, 0.75 for class II cement, and 1.15 for class III cement; *f*_ck28_ is the cubic compressive strength of concrete for 28 d, MPa.

The creep coefficient of the GL2000 model is computed from the following expression [36]
(13)$$\varphi \left(t,t_{0}\right)=\varphi \left(t_{c}\right)\left[\left\{\frac{2{\left(t-t_{0}\right)}^{0.3}}{{\left(t-t_{0}\right)}^{0.3}+14}\right\}+\sqrt{\frac{7(t-t_{0})}{t_{0}(t-t_{0}+7)}}+2.5(1-1.086h^{2})\sqrt{\frac{\left(t-t_{0}\right)}{(t-t_{0})+0.15{\left(V/S\right)}^{2}}}\right]$$
(14)$$\varphi \left(t_{c}\right)=\left\{ \begin{array}{l} 1,\;\;\;\;\;\;\;\;\;\;\;\;\;\;\;\;\;\;\;\;\;\;\;\;\;\;\;\;\;\;\;\;\;\;\;\;\;t_{0}=t_{c}\\ \sqrt{1-{\left(\frac{t_{0}-t_{c}}{t_{0}-t_{c}+0.15{\left(V/S\right)}^{2}}\right)}^{0.5}},\;\;\;\;\;t_{0}>t_{c} \end{array}\right.$$
where *t*_0_ is the age of concrete at loading time, d.

### 2.3. Shrinkage and Creep Models of CA Mortar

CA mortar is a composite material consisting of cement, fine sand, emulsified asphalt, additives, and so on. It is similar to cement mortar and concrete that the shrinkage of CA mortar includes the chemical shrinkage caused by the hydration of Portland cement and the drying shrinkage caused by the exchange of moisture with air [37,38,39]. However, there is little research on the shrinkage model of CA mortar. The research of Ref. [40] showed that the shrinkage of CA mortar is mainly drying shrinkage, and the internal relative humidity of CA mortar has a linear relationship with drying shrinkage.

Since the upper and lower sides of the CA mortar cover the track plate and the base plate, respectively, no moisture exchange with the atmosphere occurs. Meanwhile, the CA mortar is continuous along the longitudinal direction, so only two sides are exposed to the atmosphere. Therefore, it is assumed to be a one-dimensional humidity transfer case. Based on Fick’s second law, the relationship between the internal humidity of CA mortar and time was established. Then, the shrinkage strain of CA mortar was obtained. When only considering the water exchange between CA mortar and the environment, the one-dimensional humidity diffusion Equation is expressed as
(15)$$\frac{\partial h}{\partial t}=\frac{\partial }{\partial x}\left(D\frac{\partial h}{\partial x}\right)$$
where *h* is the internal humidity of CA mortar; *D* is the humidity diffusion coefficient, m^2^/d; *x* is the distance from the surface of the test piece, m.

In Equation (15), *D* is expressed in the CEB-FIP Model Code 1990 [41].
(16)$$D=\frac{D_{1,0}}{f_{\mathrm{ck}}/f_{\mathrm{ck}0}}\left[\alpha_{0}+\frac{1-\alpha_{0}}{1+{\left[\left(1-h\right)/\left(1-h_{c}\right)\right]}^{N}}\right]$$
(17)$$f_{\mathrm{ck}}=f_{\mathrm{cm}}-8$$
where *D*_1,0_ = 8.64 × 10^−5^ m^2^/d; *f*_ck0_ = 10 MPa; *f*_cm_ is the average compressive strength, MPa; *f*_ck_ is the characteristic value of compressive strength, MPa; *N* = 15; *α*_0_ = 0.05; *h_c_* = 0.8.

According to Equations (15) and (16), the internal relative humidity of CA mortar can be obtained by the finite difference method. The grid of the finite difference method is shown in Figure 3. The solution procedure of the finite difference method is as follows:
(18)$$\frac{\partial h}{\partial t}=D\frac{\partial^{2}h}{\partial x^{2}}\;\;\;\;0\leq x\leq 1$$
(19)$$\frac{\partial h}{\partial t}\left(x_{i},t_{k}\right)\approx \frac{h_{i}^{k+1}-h_{i}^{k}}{\Delta t}$$
(20)$$\frac{\partial^{2}h}{\partial x^{2}}\left(x_{i},t_{k}\right)\approx \frac{h_{i-1}^{k}-2h_{i}^{k}+h_{i+1}^{k}}{\Delta x^{2}}$$
where Δ*x* is the incremental displacement; Δ*t* is the incremental time; $h_{i}^{k}$ is the internal humidity of CA mortar at *x_i_* when time is *t_k_*.

Based on Equations (18)–(20), the following recurrence relations can be obtained.
(21)$$h_{i}^{k+1}=D\left[\frac{1}{D}h_{i}^{k}+\frac{\Delta t}{\Delta x^{2}}\left(h_{i-1}^{k}-2h_{i}^{k}+h_{i+1}^{k}\right)\right]$$
(22)$$r=\frac{\Delta t}{\Delta x^{2}}$$
(23)$$h_{i}^{k+1}=D\left[\left(\frac{1}{D}-2r\right)h_{i}^{k}+r\left(h_{i-1}^{k}+h_{i+1}^{k}\right)\right]$$

Equation (16) shows that the humidity diffusion coefficient *D* is a variable related to the relative humidity *h*. Therefore, *D* is taken as the mean value.
(24)$$D_{i}^{k}=\frac{1}{2}\frac{D_{1,0}}{f_{\mathrm{ck}}/f_{\mathrm{ck}0}}\left[2\alpha_{0}+\frac{1-\alpha_{0}}{1+{\left[\left(1-h_{i-1}^{k}\right)/\left(1-h_{c}\right)\right]}^{N}}+\frac{1-\alpha_{0}}{1+{\left[\left(1-h_{i+1}^{k}\right)/\left(1-h_{c}\right)\right]}^{N}}\right]$$

In addition, the increment of shrinkage strain keeps a linear relationship with the increment of relative humidity [42]. Therefore, the shrinkage strain can be obtained.
(25)$$\Delta \varepsilon_{sh}=\Delta h\cdot \alpha_{sh}$$
where $\Delta \varepsilon_{sh}$ is the increment of dry shrinkage strain; Δ*h* is the relative humidity increment; *α*_*sh*_ is the dry shrinkage coefficient, recommended value of 0.0015 [43].

The ambient relative humidity was 65% in Ref. [39], and the final shrinkage value can be obtained by an exponential function, as shown in Figure 4.

From Figure 4, the final shrinkage value of CA mortar is about 613 × 10^−6^. According to Equation (25), the ultimate drying shrinkage of CA mortar is 525 × 10^−6^. The latter is 85.6% of the former. However, Equation (25) does not include the chemical shrinkage of CA mortar, so it is reasonable to use Equation (25) to calculate the dry shrinkage strain.

According to the results of Ref. [44], the creep calculation model of CA mortar is as follows:(26)$$\varepsilon =\frac{\sigma_{0}}{E_{1}}\left[1+\varphi \left(t\right)\right]$$
(27)φ(t)=1.842[1−exp(−0.00455t)]+2.376[1−exp(−0.04036t)]

It can be seen that Equation (27) is consistent with Equation (3), so the calculation of the creep effect of CA mortar is consistent with that of concrete.

### 2.4. Verification of the Secondary Development of ABAQUS

The concrete prism models with the dimension of 100 mm×100 mm×400 mm under the dead load and the time-dependent loading were established, respectively. Further, the theoretical displacement and the finite element displacement of the top surface of the model caused by shrinkage and creep were analyzed. The concrete prism models are shown in Figure 5.

In Figure 5, the bottom surface of the model is fixed, and the spring is set on the top surface. In addition, the spring stiffness was changed to simulate different load conditions. Under the action of dead load, the spring stiffness is 0. When the spring stiffness is 16,000 N/mm, it is a time-dependent loading effect.

The theoretical value was calculated by the age-adjusted effective modulus method. The age-adjusted effective modulus method is actually an algebraic form of Equation (1) considering the initial elastic deformation.
(28)$$\varepsilon \left(t\right)=\frac{\sigma \left(t_{0}\right)}{E_{c}}\left[1+\varphi \left(t,t_{0}\right)\right]+\frac{\sigma \left(t\right)-\sigma \left(t_{0}\right)}{E_{c}}\left[1+\chi \varphi \left(t,t_{0}\right)\right]+\varepsilon_{sh}\left(t\right)$$

Based on Equation (28), the theoretical calculation expressions can be obtained.
(29)$$\omega_{\mathrm{dead}\;\mathrm{load}}=\frac{\sigma_{0}}{E_{c}}h_{c}\left[1+\varphi \left(t,t_{0}\right)\right]+\varepsilon_{sh}h_{c}$$
(30)$$\omega_{\mathrm{time}-\mathrm{dependent}\;\mathrm{loading}}=\sigma_{0}\frac{E_{c}A}{E_{c}A+kh_{c}}\frac{\left[1+\chi \varphi \left(t,t_{0}\right)\right]+\left(1-\chi \right)\varphi \left(t,t_{0}\right)}{E_{c}A+\left(1-\chi \right)\varphi \left(t,t_{0}\right)kh_{c}}Ah_{c}+\varepsilon_{sh}\frac{E_{c}Ah_{c}}{E_{c}A+\left(1-\chi \right)kh_{c}}$$
where $\sigma_{0}$ is the initial load; *A* is the cross-sectional area of the prism model; *E_c_* is the elastic modulus of concrete; *h_c_* is the height of prism model; *k* is the spring stiffness; $\varphi \left(t,t_{0}\right)$ is creep coefficient; *ε_sh_* is the shrinkage strain.

In Equations (29) and (30), the creep and shrinkage of concrete are predicted by the GL2000 model. The ambient relative humidity is 70%, and the loading age is 3 d. The displacement result is shown in Figure 6.

It can be seen from Figure 6 that there is a very good matching between the theoretical results and the finite element values. In fact, the theory of calculation of the theoretical values is the same as that used for the finite element model. Therefore, the numerical results should coincide with the theoretical values. It can be concluded that ABAQUS is reliable for the analysis of shrinkage and creep effects of concrete.

## 3. Finite Element Modeling

### 3.1. Information of Model

The prefabricated box girder was connected with the base plate through the shear cogging at the girder end. For the remaining areas, only the sliding layer was arranged, which can only bear compression. Due to the limited constraint of the box girder on the ballastless track, it was only used as the vertical support structure of the ballastless track. Therefore, the shrinkage and creep of the box girder are not considered in the model. Furthermore, to simplify the model, the box girder was replaced by a plate with a thickness of 0.3 m (the thickness of the top plate of the box girder). The grounding spring was set at the bottom of the plate. The structural model of the CRTS II slab ballastless track on the bridge is shown in Figure 7.

From Figure 7, one can see that the track slab is a prestressed concrete structure, so the concrete shrinkage and creep should be considered. However, the base slab is a reinforced concrete member, and only its shrinkage deformation is calculated. Because the creep rate of CA mortar is generally much higher than that of ordinary concrete [44], its shrinkage and creep are all considered. Obviously, when the deformation of each layer is uncoordinated, additional stresses inevitably occur between the layers of the ballastless track.

Since CRTS II slab ballastless track is a continuous longitudinal system, to weaken the influence of boundary conditions [45], a ballastless track model with a length of 26 m (that is, the length of four track slabs) was established. Figure 8 is the planar graph of the ballastless track model.

In order to explore the restraint effect of the shear cogging on the ballastless track, section A and section B in Figure 8 (i.e., with and without the shear cogging) were taken as the analysis positions of interlayer stress.

The dimensions of each component of the ballastless track model are shown in Table 1.

In the proposed finite element model, the friction coefficient of the sliding layer was 0.2 [45], and the contact stiffness was 1.8 MPa/mm according to the “Temporary technical conditions for the sliding layer of CRTS II slab ballastless track of passenger dedicated railway”.

The vertical and transverse stiffness of the shear cogging were 2.0 × 10^5^ kN/mm and 300 kN/mm, respectively, and its longitudinal stiffness was close to rigidity [46]. The limit values of vertical and lateral strength of the shear cogging are taken as 206 kN and 119 kN, respectively [46]. The COH3D8 cohesion element was to simulate the interface between CA mortar and the track slab. The relationship between the interface stress and the displacement was bilinear [45]. The cohesion model parameters are shown in Table 2 [47].

The material parameters of the CRTS II slab ballastless track are listed in Table 3 [45], and the elements used for each component of the ballastless track model are shown in Table 4.

For reasons of better accuracy and efficiency, quadrilateral elements are preferred for two-dimensional meshes and hexahedral elements for three-dimensional meshes [48]. Therefore, hexahedral elements are adopted. The model grid of the ballastless track is shown in Figure 9.

In addition, the prestressed tendons of the track slab were tensioned at 7 d, and the ambient relative humidity in the shrinkage and creep model was 70%. The initial relative humidity of CA mortar is 100%, and the average strength is 11.35 MPa [49].

### 3.2. Grid of the Finite Model

The grid size is a key parameter that affects the accuracy of the finite element results. In general, the calculation accuracy increases with the decrease in the grid size. However, the small size also leads to a longer computation time. Therefore, a reasonable grid size should be selected with full consideration of calculation accuracy and efficiency.

A simplified model of ballastless track, i.e., Figure 10, was established in accordance with Figure 7 to determine the grid size. The grid size of the model is 7.5 cm, 5 cm, and 2.5 cm, respectively. The boundary conditions and material parameters are consistent with Section 3.1. Two approaches to reducing the computational effort are as follows.

(1)The longitudinal length of the model is reduced to 2 m;(2)Transverse loads are arranged on the side of the track slab to simulate the difference in shrinkage and creep of each layer of ballastless track.

The maximum transverse shear stress at the interface between the track slab and CA mortar is shown in Figure 11.

It can be observed in Figure 11 that when the grid size is 5 cm and 2.5 cm, respectively, the difference between the two interface stresses is small. This shows that the latter has little improvement in computational accuracy compared to the former, but the number of elements of the latter is eight times higher than that of the former. The calculation time is positively correlated with the number of elements. Therefore, the grid size of the finite model is taken as 5 cm.

## 4. Results and Discussion

The difference of shrinkage and creep increases with the age difference of each layer of the ballastless track. Therefore, the concrete age of the track slab is a critical factor affecting the cooperative deformation of the ballastless track. It is assumed that the concrete age of the track slab is 60 d, 120 d, 180 d, 270 d, and 360 d (i.e., the AOTSL) based on the construction time and storage time of the track slab. The additional force of the shear cogging and the interlayer stress of the ballastless track within 360 d after the track slab laying were analyzed.

### 4.1. Relative Humidity and Shrinkage Strain of CA Mortar

According to Equations (21)–(25), the internal relative humidity and dry shrinkage strain of CA mortar is shown in Figure 12 and Figure 13, respectively.

It can be seen from Figure 12 that the rate of change in relative humidity inside CA mortar decreases with the depth from the surface during the drying process. At the same time, as the drying time increases, the relative humidity inside CA mortar gradually decreases, and the drying depth increases. It can be observed in Figure 13 that the shrinkage strain of CA mortar gradually increases with time but decreases with the increasing depth from the surface.

### 4.2. Model Validation

Stresses of the sliding layer and interface stresses between CA mortar and the track slab at day 0 are shown in Figure 14 and Figure 15, respectively.

From Figure 14, it can be seen that the normal stress ranges from 4.79 KPa to 11.89 KPa, and the transverse shear stress is negligible. The theoretical normal stress of 9.96 KPa is in the range of finite element values. The theoretical transverse shear stress is 0 KPa, which is very close to the finite element value.

It can be seen from Figure 15 that the normal stress ranges from 4.34 KPa to 6.28 KPa, and the transverse shear stress ranges from −3.20 KPa to 3.25 KPa. The theoretical normal stress of 5.10 KPa is in the range of finite element values. Although the theoretical transverse shear stress of 0 KPa is quite different from the finite element value, it is much smaller than the interface strength of 956 KPa.

Therefore, the proposed finite element model basically conforms to the actual initial stress of the structure. It can be concluded that the finite element model of the CRTS II slab ballastless track can be used for subsequent structural analysis.

### 4.3. Normal Stress of the Sliding Layer

When the lateral deformation of each layer of the ballastless track is not coordinated, the stress redistribution of the contact surface (i.e., the sliding layer) between the base plate and the box girder is induced. If there is a void in the sliding layer, the base plate and the girder surface have a mutual flapping effect under the live load, which adversely affect their service. Figure 16 and Figure 17 are the normal stress of the sliding layer at different AOTSL (the stress is positive under compression). Figure 18 and Figure 19 illustrate the variation in the gap length of the sliding layer with time at different AOTSL.

It can be observed in Figure 16 and Figure 17 at day 0 that the normal stress is negative, indicating that the sliding layer is in compression after the laying of the track slab. Due to the shrinkage and creep of the ballastless track, the normal stress of the sliding layer is redistributed along the transverse direction. At the AOTSL of 60 d, the normal stress of the sliding layer at the middle of the slab decreases firstly and then increases, while the stress at the slab edge increases firstly and then decreases to 0, which indicates that the sliding layer appears a gap at the slab edge. At the AOTSL of 360 d, the normal stress of the sliding layer at the slab edge gradually increases, while the stress at the middle of the slab gradually decreases to 0, which reveals that the sliding layer appears a gap at the middle of the slab. Moreover, the shear cogging is able to reduce the gap length to a certain extent.

From Figure 18, at the AOTSL of 60 d, it is observed that the sliding layer at the middle of the slab is always in compression. When the AOTSL is between 120 d and 180 d, the sliding layer at the middle of the slab appears a gap with time, but eventually, the gap length decreases to 0. When the AOTSL is between 270 d and 360 d, the gap length of the sliding layer increases sharply with time before 60 d and then tends to be flat. Moreover, when the AOTSL is 360 d, for the cases of with and without the shear cogging effects, the gap length of the sliding layer at the middle of the slab is 2.12 m and 1.75 m at day 360, respectively. Thus, the shear cogging reduces the gap length of the sliding layer at the middle of the slab. In addition, the gap length of the sliding layer at the middle of the slab increases with the increase in AOTSL.

It can be seen from Figure 19 that, without the shear cogging, a gap gradually appears with time in the sliding layer at the slab edge only when the AOTSL is 60 d. When considering the effect of the shear cogging, a gap gradually appears with time in the sliding layer at the slab edge when the AOTSL is less than 180 d. Conversely, the sliding layer at the slab edge is always at the compression state when the AOTSL is more than 180 d. In detail, at the AOTSL of 60 d, the gap length of the sliding layer at the slab is edge 0.73 m and 0.57 m at day 360 with and without the shear cogging, respectively. In addition, although the shear cogging reduces the gap length of the sliding layer at the slab edge, the sliding layer at the slab edge is more prone to produce a gap. Moreover, the gap length of the sliding layer at the slab edge decreases with the increase in AOTSL. Further, by comparing Figure 18 and Figure 19, a conclusion can be drawn that when the AOTSL ranges from 120 d to 180 d, the sliding layer has a relatively small gap length.

### 4.4. Interfacial Stress between CA Mortar and the Track Slab

Similarly, the redistribution of interface stress has occurred at the interface between CA mortar and the track slab due to the uncoordinated deformation of each layer of the ballastless track, which results in additional stresses. Figure 20 and Figure 21 show the distribution of the transverse interface shear stress at the interface between the track slab and CA mortar with and without the shear cogging. In Figure 22, the time-dependent law of the transverse interface shear stress at the track slab edge at different AOTSL is presented.

It can be seen from Figure 20 and Figure 21 that the interface transverse shear stress gradually emerged due to the uneven lateral shrinkage and creep deformation of the track slab and CA mortar. In detail, the interfacial stress is negligible at the middle of the slab and increases gradually from the middle to the edge of the slab. At the AOTSL of 60 d, the interfacial stress reaches the peak value at the slab edge and about 0.1 m away from the slab edge with time. At the AOTSL of 360 d, the interfacial stress reaches the maximum only at the slab edge. When comparing Figure 20 with Figure 21, it can be seen that the effect of the shear cogging on interface stress is negligible.

It can be seen from Figure 22 that the interfacial stress at the slab edge increases sharply at first and then decreases gradually with time. In addition, the interfacial stress increases with the increase in AOTSL. Moreover, the time of the peak stress appearance increases gradually with the increase in AOTSL. At the AOTSL of 60 d, the interfacial stress reaches the peak value of 0.440 MPa at 28 d with and without the shear cogging. When the AOTSL is 360 d, the interfacial stress reaches the peak value at 180 d, which is 0.676 MPa and 0.642 MPa for the cases of with and without the shear cogging effects, respectively, reaching 70.7% and 67.2% of the interfacial shear strength. Thus, it can be concluded that the effect of the shear cogging on interfacial stress can be ignored. Furthermore, to reduce the additional interface stress caused by the shrinkage and creep effects of the ballastless track, the track slab should be laid as soon as possible.

### 4.5. Additional Forces of the Shear Cogging

The shear cogging is a key connector to transfer the longitudinal force and limit the displacement of the base plate. Hence, it is of great significance to explore the force of the shear cogging. Figure 23 and Figure 24 show the distribution of vertical force and lateral force of the shear cogging at different AOTSL. Figure 25 is the time-varying diagram of forces at the edge and middle of the slab. The force is positive in tension and negative in compression.

As seen from Figure 23 and Figure 24, it can be seen that the vertical and lateral additional forces are gradually generated due to the shrinkage and creep of the ballastless track. The peak value of vertical additional force appears at the middle and edge of the slab. However, the lateral additional force is the largest at the slab edge and gradually decreases towards the middle of the slab. The additional force of the shear cogging is much less than its strength limit. At the AOTSL of 60 d, as time increases, the shear cogging is under vertical tension at the slab edge and compression at the middle of the slab. When the AOTSL is 360 d, the shear cogging is under vertical compression at the slab edge and tension at the middle of the slab over time. In addition, except for the middle of the slab, the shear cogging is always in lateral tension under different AOTSL.

It is found in Figure 25 that the vertical force of the shear cogging shows great discrepancies at different AOTSL. In detail, when the AOTSL is less than 180 d, the vertical force of the shear cogging at the slab edge changes from negative to positive with time, which indicates that the shear cogging changes from vertical compression to vertical tension. When the AOTSL is more than 180 d, the vertical force at the slab edge is always negative, indicating that the shear cogging at the slab edge is constantly in vertical compression. In the middle of the slab, the vertical force firstly increases and then decreases to negative at the AOTSL less than 180 d, which indicates that the shear cogging is in vertical compression. When the AOTSL is more than 180 d, the vertical force increases gradually with time, and the shear cogging at the middle of the slab is finally in vertical tension. In addition, the lateral force of the shear cogging at the slab edge increases with time but decreases slightly with the increase in AOTSL. Furthermore, according to the additional force of the shear cogging, the AOTSL should preferably range from 120 d to 180 d.

## 5. Conclusions

In this work, a finite model of CRTS II slab ballastless track was established to analyze interlaminar stress of the ballastless track and the additional force of the shear cogging under the effect of shrinkage and creep of the ballastless track. Conclusions can be drawn as follows:(1)Under the effect of shrinkage and creep of the ballastless track, the normal stress redistribution of the sliding layer emerges. Further, a gap of the sliding layer occurs at the middle or edge of the slab. The shear cogging can slightly reduce the gap length, but the sliding layer at the slab edge is more prone to produce a gap;(2)The shrinkage and creep of the ballastless track cause the additional transverse interface shear stress between the track slab and CA mortar, which is the largest at the slab edge and gradually increases with the increase in AOTSL. At the AOTSL of 360 d, the maximum stress is 0.676 MPa and 0.642 MPa with and without the shear cogging, respectively, reaching 70.7% and 67.2% of the interface shear strength, but no interface damage is caused;(3)The peak value of the vertical additional force appears at the middle and edge of the slab. In comparison, the lateral additional force reaches the maximum at the slab edge and decreases gradually towards the middle of the slab. Additional forces are not yet sufficient to cause damage to the shear cogging;(4)Through a comprehensive analysis of the gap length of the sliding layer, the interface stress, and the additional force of the shear cogging, the suggested AOTSL ranges 120 d from 180 d.

## Figures and Tables

**Figure 1 materials-15-02480-f001:**
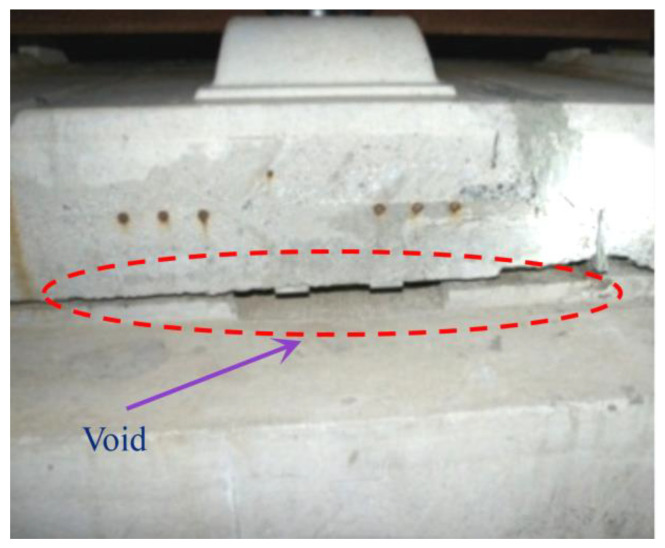
The void in the filling layer.

**Figure 2 materials-15-02480-f002:**
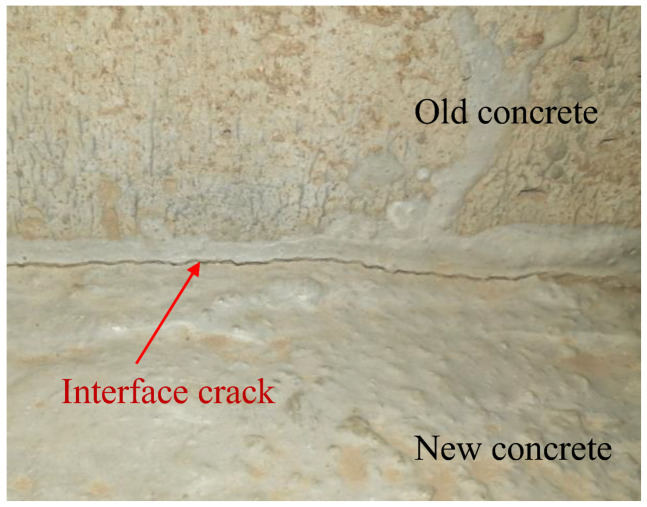
Interface crack of the new–old concrete.

**Figure 3 materials-15-02480-f003:**
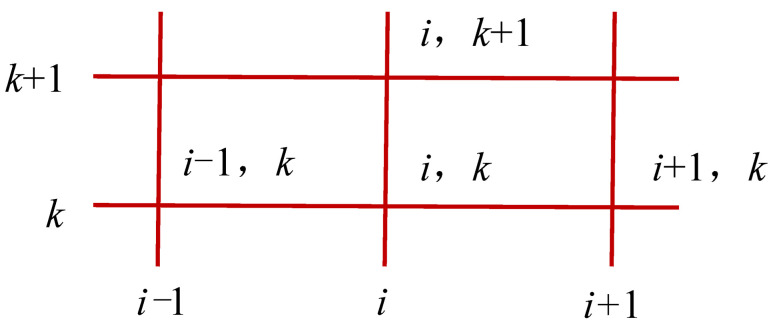
Grid of the finite difference method.

**Figure 4 materials-15-02480-f004:**
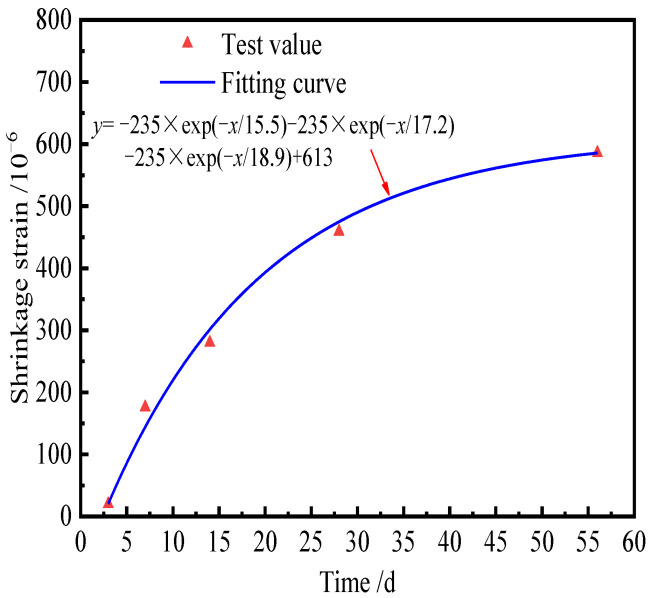
Shrinkage strain of CA mortar.

**Figure 5 materials-15-02480-f005:**
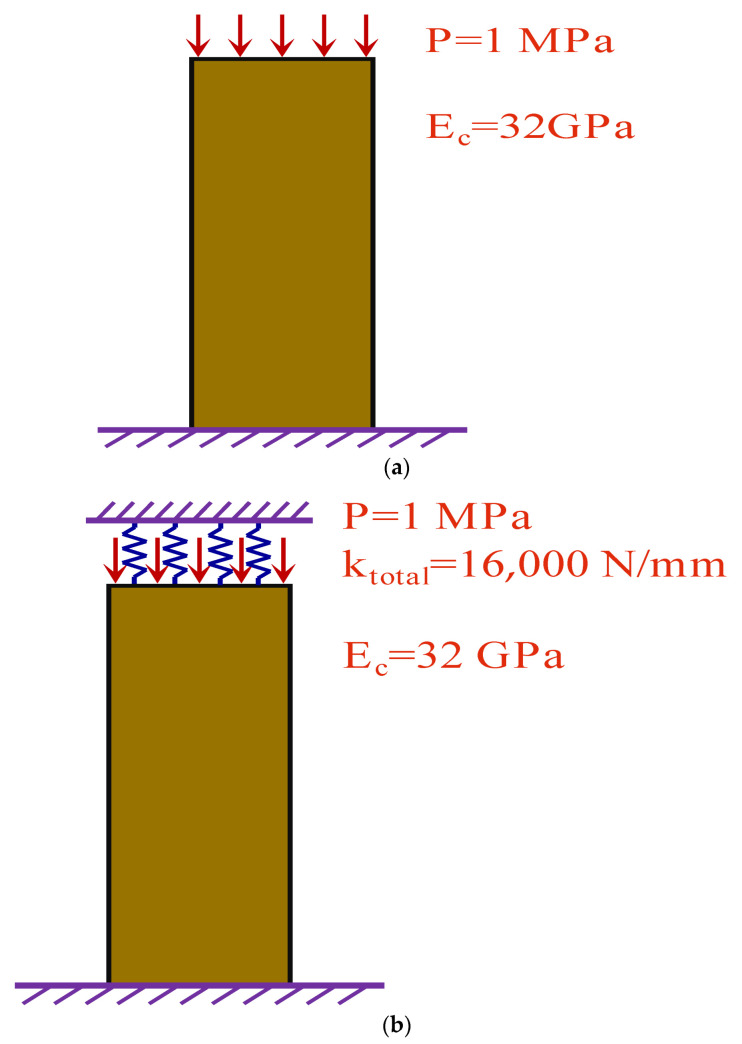
Model of the concrete prism under load. (**a**) Dead load. (**b**) Time-dependent loading.

**Figure 6 materials-15-02480-f006:**
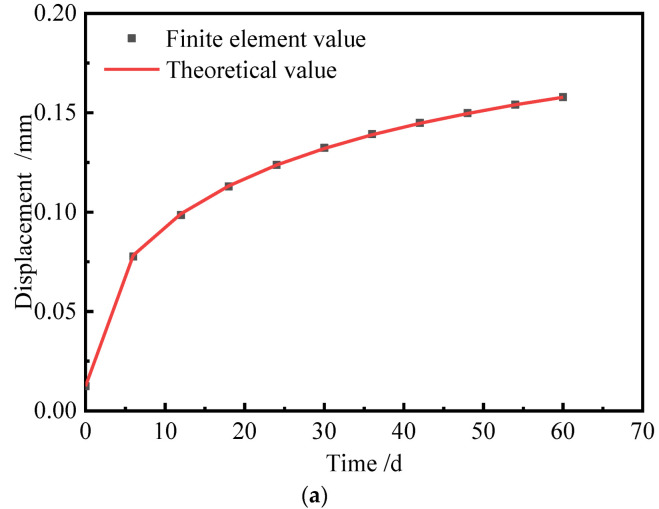
Displacement of the top surface of the model. (**a**) Dead load. (**b**) Time-dependent load.

**Figure 7 materials-15-02480-f007:**
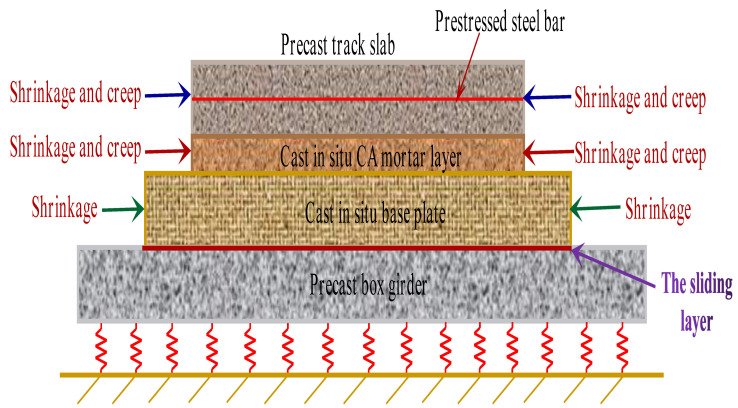
Model of CRTS II slab ballastless track.

**Figure 8 materials-15-02480-f008:**
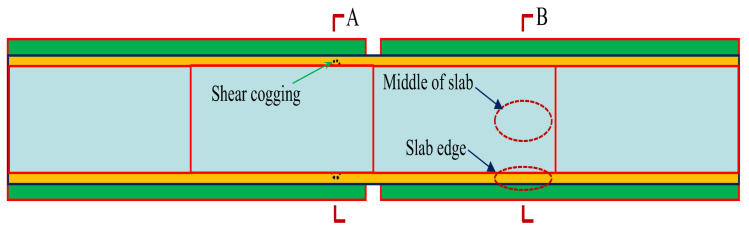
Planar graph of the ballastless track model.

**Figure 9 materials-15-02480-f009:**
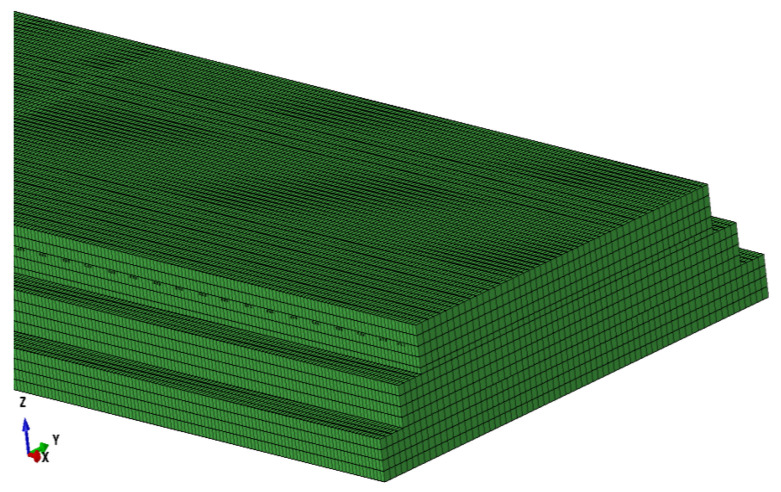
Grid diagram of the ballastless track.

**Figure 10 materials-15-02480-f010:**
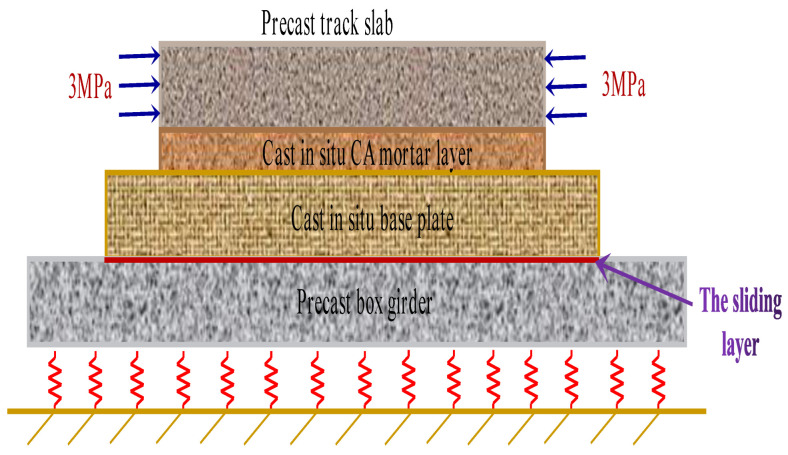
Simplified model.

**Figure 11 materials-15-02480-f011:**
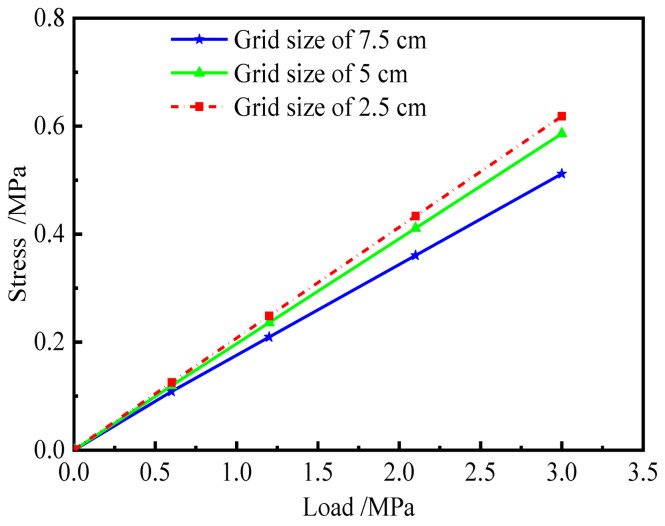
Interface shear stress.

**Figure 12 materials-15-02480-f012:**
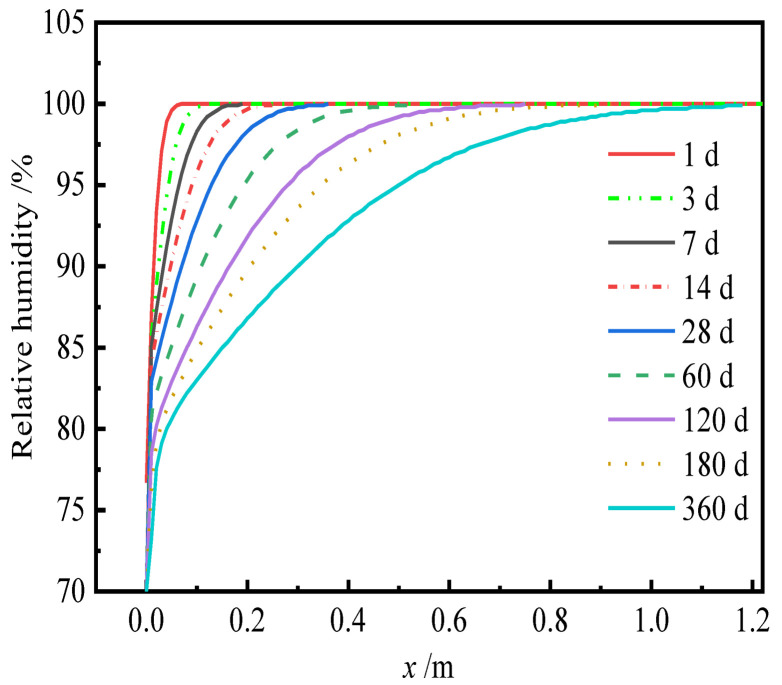
Relative humidity of CA mortar.

**Figure 13 materials-15-02480-f013:**
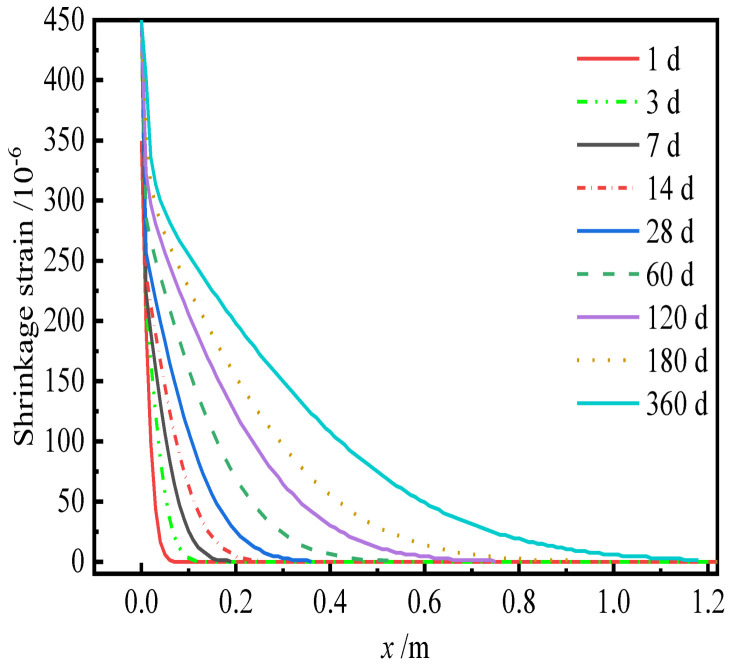
Dry shrinkage strain of CA mortar.

**Figure 14 materials-15-02480-f014:**
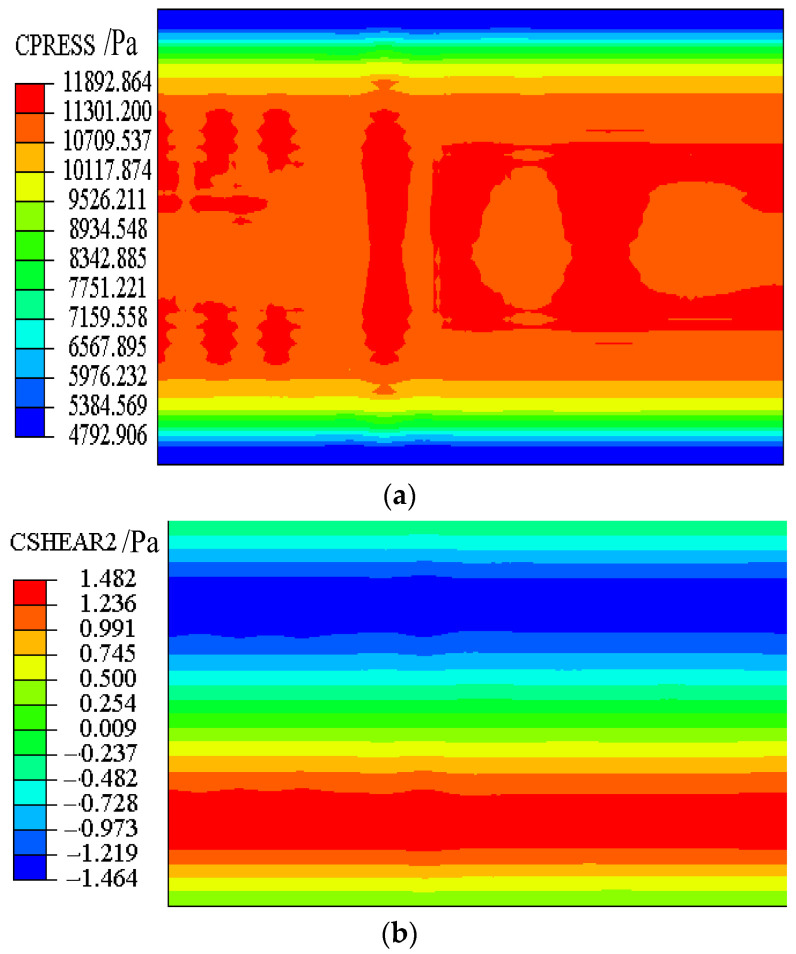
Stress of the sliding layer. (**a**) Normal stress. (**b**) Transverse shear stress.

**Figure 15 materials-15-02480-f015:**
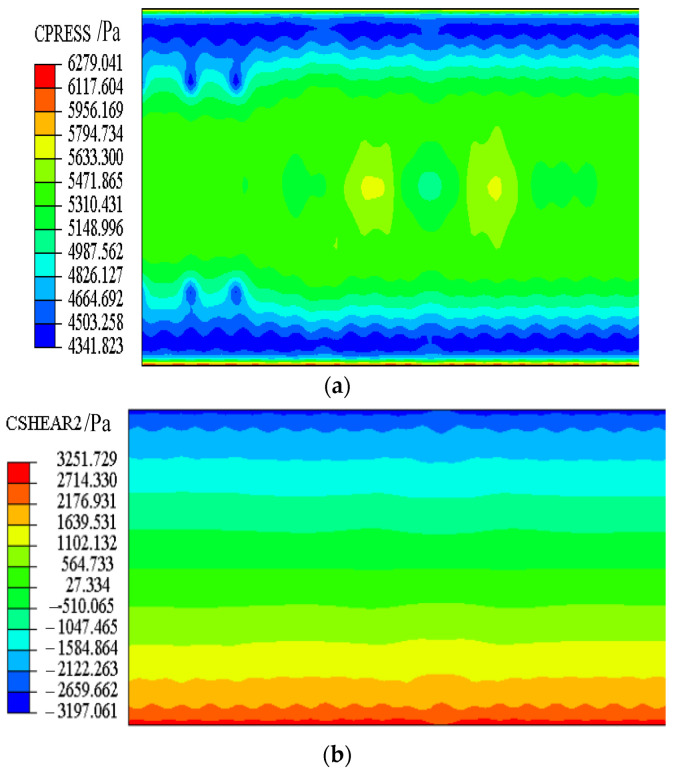
Interface stress. (**a**) Normal stress. (**b**) Transverse shear stress.

**Figure 16 materials-15-02480-f016:**
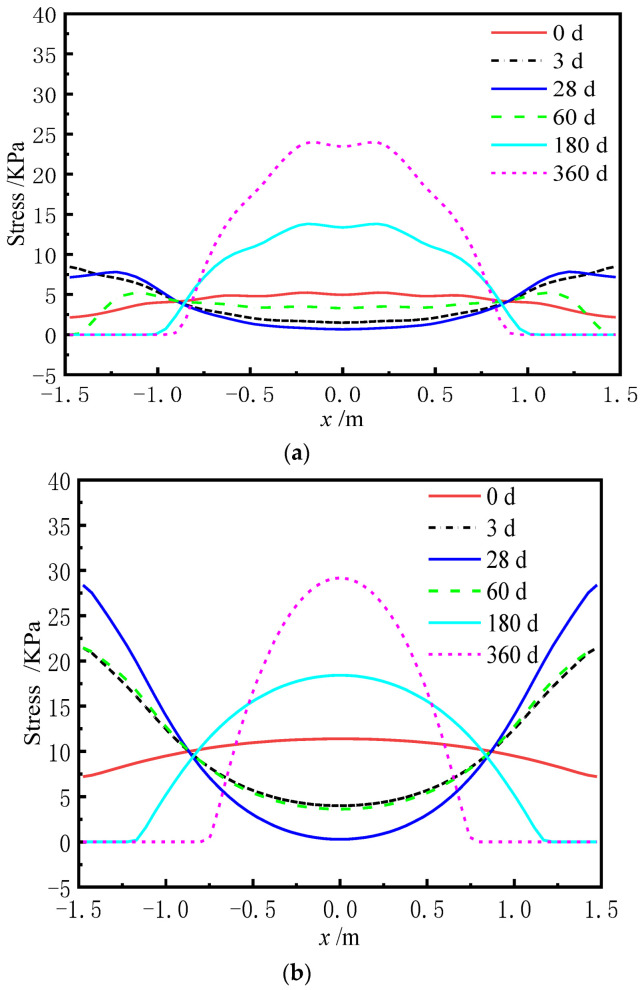
Normal stress of the sliding layer at the AOTSL of 60 d. (**a**) With the shear cogging. (**b**) Without the shear cogging.

**Figure 17 materials-15-02480-f017:**
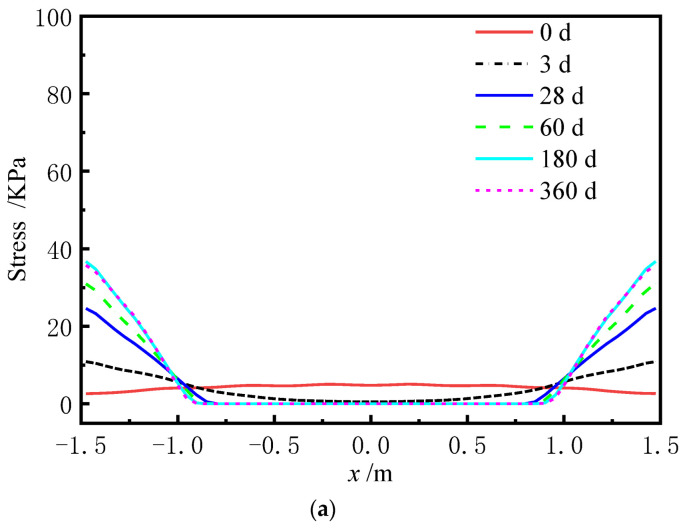
Normal stress of the sliding layer at the AOTSL of 360 d. (**a**) With the shear cogging. (**b**) Without the shear cogging.

**Figure 18 materials-15-02480-f018:**
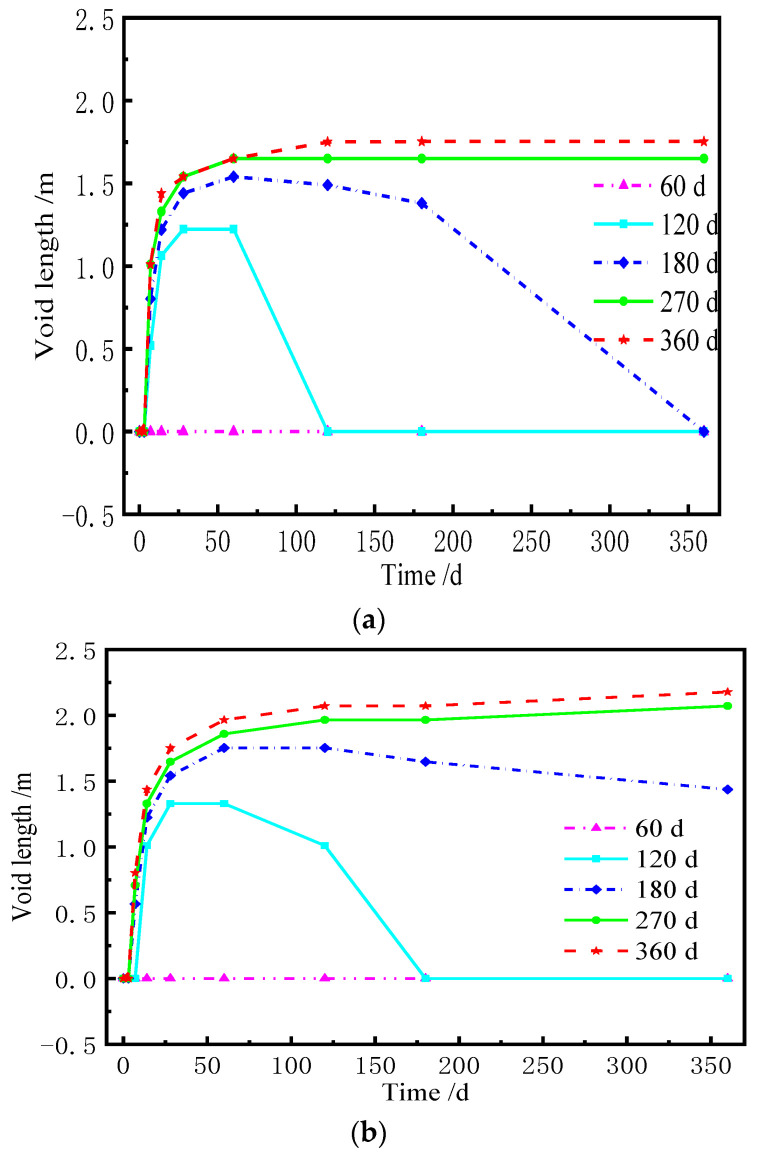
Gap length of the sliding layer at the middle of the slab. (**a**) With the shear cogging. (**b**) Without the shear cogging.

**Figure 19 materials-15-02480-f019:**
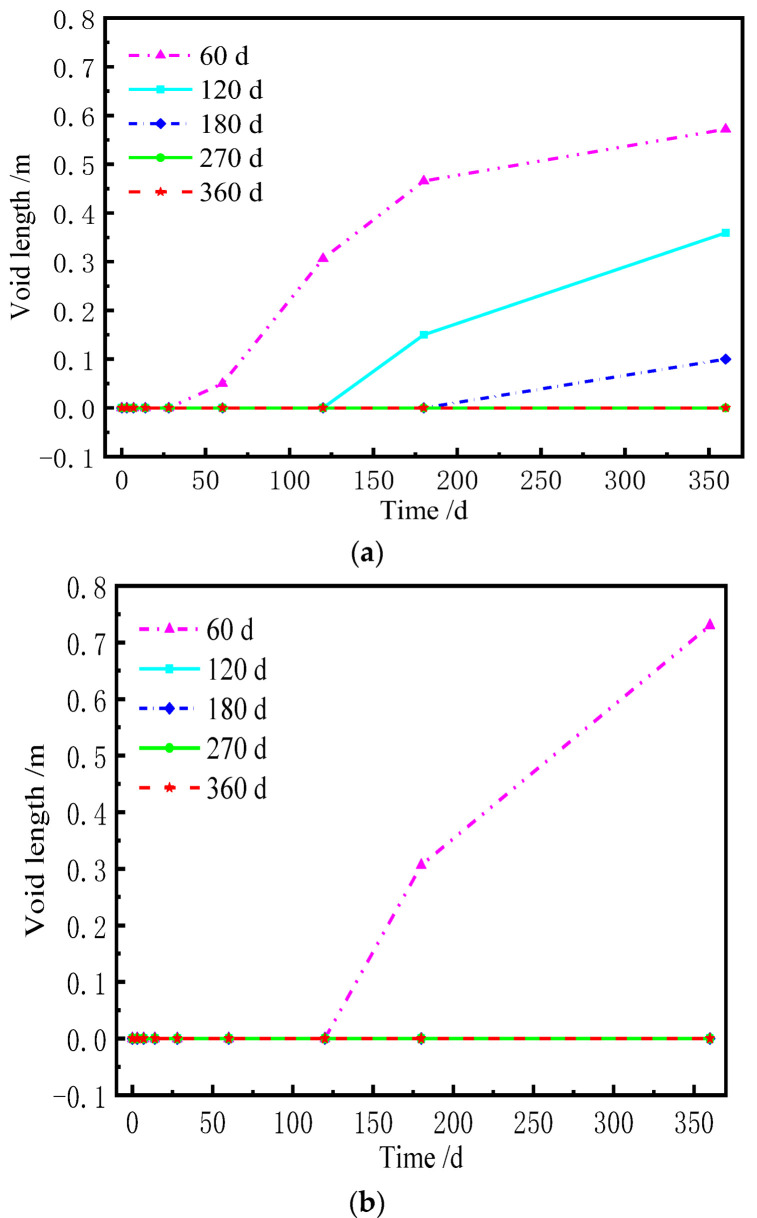
Gap length of the sliding layer at the slab edge. (**a**) With the shear cogging. (**b**) Without the shear cogging.

**Figure 20 materials-15-02480-f020:**
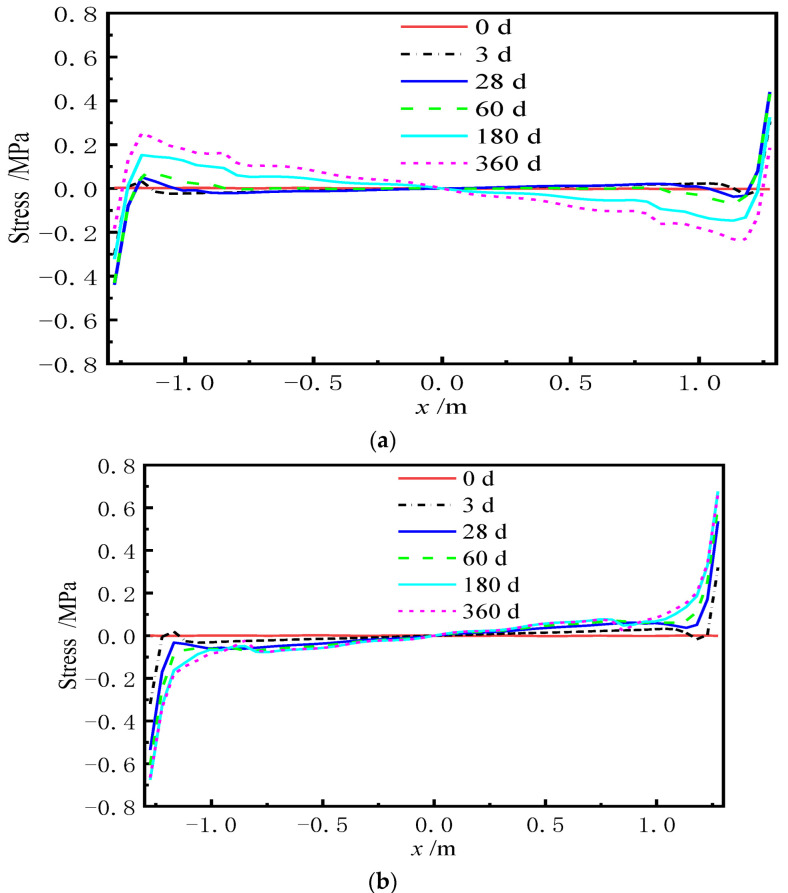
Interfacial stress with the shear cogging. (**a**) AOTSL of 60 d. (**b**) AOTSL of 360 d.

**Figure 21 materials-15-02480-f021:**
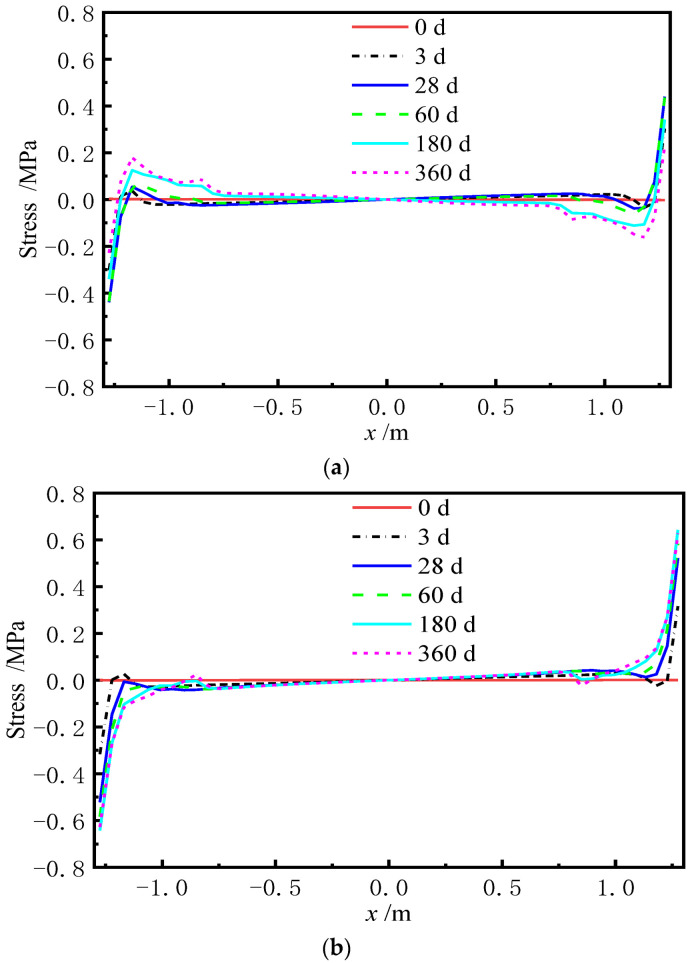
Interfacial stress without the shear cogging. (**a**) AOTSL of 60 d. (**b**) AOTSL of 360 d.

**Figure 22 materials-15-02480-f022:**
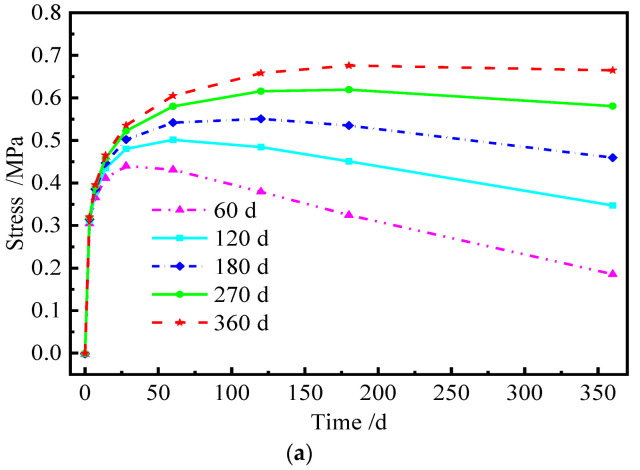
Time-varying interfacial stress. (**a**) With the shear cogging. (**b**) Without the shear cogging.

**Figure 23 materials-15-02480-f023:**
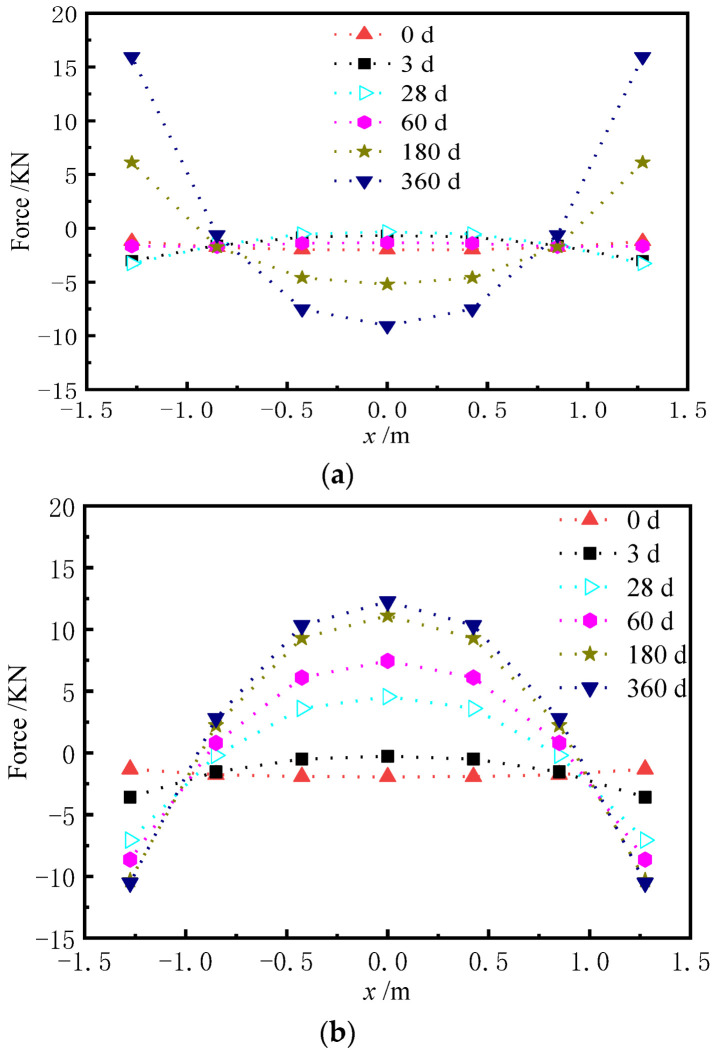
Vertical force of the shear cogging. (**a**) AOTSL of 60 d. (**b**) AOTSL of 360 d.

**Figure 24 materials-15-02480-f024:**
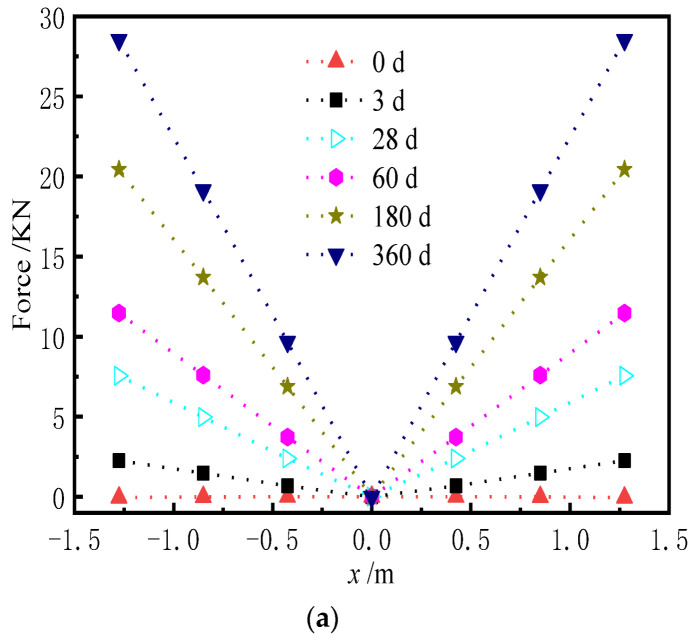
Lateral force of the shear cogging. (**a**) AOTSL of 60 d. (**b**) AOTSL of 360 d.

**Figure 25 materials-15-02480-f025:**
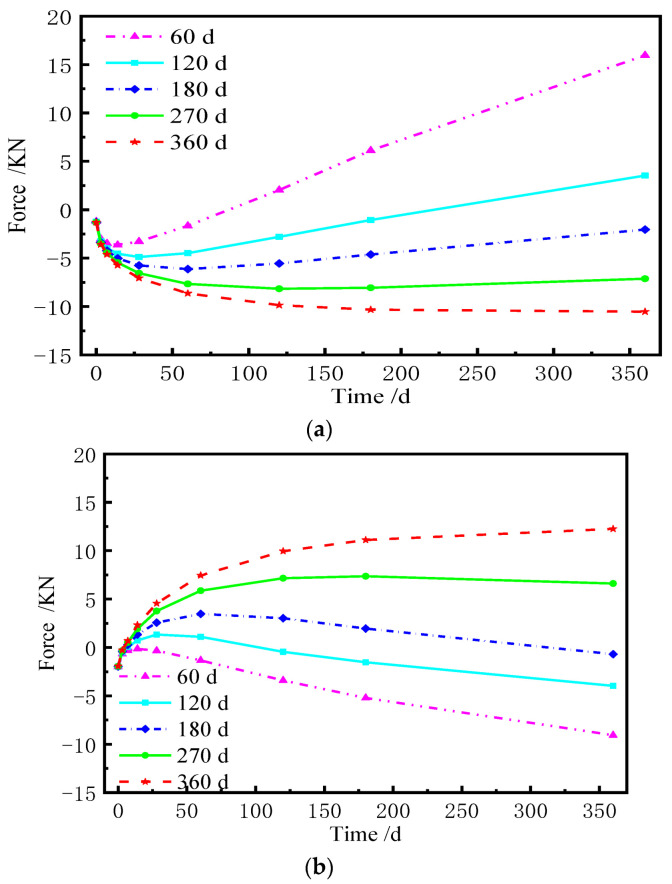
Force time-varying graph of the shear cogging. (**a**) Vertical force at the slab edge. (**b**) Vertical force at the middle of slab. (**c**) Lateral force at the slab edge.

**Table 1 materials-15-02480-t001:** Dimensions of each component.

Component	Width/mm	Thickness/mm	Length/m
Track slab	2550	200	4 × 6.5 = 26
Base plate	2950	200	26
CA mortar	2550	30	26

**Table 2 materials-15-02480-t002:** Cohesion model parameters.

Items	$t_{n}^{0}/\mathrm{MPa}$	$\delta_{n}^{0}/\mathrm{mm}$	$\delta_{n}^{f}/\mathrm{mm}$
Normal direction	1.792	0.0025	0.0282
Tangential direction	0.956	0.0152	0.0376

**Table 3 materials-15-02480-t003:** Material parameters of CRTS II slab ballastless track.

Materials	Modulus of Elasticity/GPa	Density/kg m^−3^	Poisson Ratio
Box girder concrete (C50)	35.5	2600	0.2
Prestressed reinforcement	195	7850	0.3
Track slab concrete (C55)	36	2600	0.2
CA mortar	7	1800	0.2
Base plate concrete (C40)	32	2500	0.2

**Table 4 materials-15-02480-t004:** Elements of finite-element model.

Parts	The Box Girder, the Track Slab, the Base Plate, and CA Mortar	The Interface between CA Mortar and the Track Slab	Rebar	The Shear Cogging
Elements	C3D8I	COH3D8	T3D2	Spring

## Data Availability

The data presented in this study are available on request from the corresponding author.

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
