# Peer review of "Numerical Investigation of Interlaminar Stress of CRTS II Slab Ballastless Track Induced by Creep and Shrinkage of Concrete"

_materials, 2022, doi:10.3390/ma15072480_

Round 1

Reviewer 1 Report

In Figure 3 the theoretical and numerical results are quasi-identical, is it possible? How can you explain that?

A validation with an experimental works or a verification and a numerical work is to be performed to check the validity of you model.

A grid sensitivity test is to be performed.

Why cubic elements are chosen and not tetrahedral?

The x, y, z coordinates are to be added to fig 6.

Why a one-dimensional humidity diffusion is employed ? is it logical for the considered dimensions?

The authors solved 3D configuration, but presented only 1D results.

The conclusion is to be reduced.

Author Response

Dear the reviewers,

Thank you very much for your careful and constructive comments on the manuscript (No. materials-1612399, title: Study on interlaminar stress of CRTS II slab ballastless track under concrete shrinkage and creep). Your comments are all valuable and very helpful for revising and improving our paper. According to your comments, we revised the manuscript and made correction which we hope meet with approval, and the modified parts were all marked in blue.

I hope you are satisfied with the revised version, however, if there is any question, we are willing to revise it again.

Yours sincerely,

Zhihui Zheng

Next section is the reply to reviewers’ comments:

Reviewer #1:

Comments and Suggestions for Authors

  1. In Figure 3 the theoretical and numerical results are quasi-identical, is it possible? How can you explain that?

(Yes. According to the reviewers’ recommendations, the corresponding explanation was added, i.e., lines from 254 to 256 and 267 to 269, which were marked in blue. In fact, the theory of calculation of the theoretical values is the same as that used for the finite element model. Therefore, the numerical results should be coincided with the theoretical values.)

  1. A validation with an experimental works or a verification and a numerical work is to be performed to check the validity of you model.

(Your recommendation is very pertinent. In this paper, a finite element model was used to study the interlaminar stresses caused by the shrinkage and creep of materials. The objective is to explore the potential damage that may be caused by the shrinkage and creep of ballastless track. At present, it is difficult to directly test the interfacial stresses. Additionally, there are no reports of experimental data or the result of a finite element model in the existing studies. Therefore, this paper only contains the finite element value. In addition, the section 4.2 was added to verify the initial stress of the structure, i.e., lines from 375 to 395, which were marked in blue. Further, the author will carry out relevant experiments in next study. )

  1. A grid sensitivity test is to be performed.

(OK. According to the reviewers’ recommendations, the section 3.2 was added to determine the grid size, i.e., lines from 331 to 355, which were marked in blue.)

  1. Why cubic elements are chosen and not tetrahedral?

(OK. According to the reviewers’ recommendations, the corresponding explanation was added, i.e., lines from 322 to 324, which were marked in blue. For reasons of better accuracy and efficiency, quadrilateral elements are preferred for two-dimensional meshes and hexahedral elements for three-dimensional meshes. This preference is clear in structural analysis and seems to also hold for other engineering disciplines.)

  1. The x, y, z coordinates are to be added to fig 6.

(OK. According to the reviewers’ recommendations, the x, y, z coordinates were added in Fig. 9.)

  1. Why a one-dimensional humidity diffusion is employed ? is it logical for the considered dimensions?

(OK. Since the upper and lower sides of the CA mortar cover the track plate and the base plate respectively, no moisture exchange with the atmosphere occurs. Meanwhile, the CA mortar is continuous along the longitudinal direction, so only two sides are exposed to the atmosphere. Therefore, it is assumed to be a one-dimensional humidity transfer problem. According to the reviewers’ recommendations, the corresponding explanations were added, i.e., lines from 188 to 192, which were marked in blue.)

  1. The authors solved 3D configuration, but presented only 1D results.

(Your recommendation is very pertinent. The interlaminar stresses are only at an interface in the vertical direction and do not vary along the vertical direction. At the same time, the interlaminar stresses vary little along the longitudinal direction because the model is constrained at both ends, only the transverse distribution of interlayer stress along the track slab is analyzed. Therefore, 1D results are presented.)

  1. The conclusion is to be reduced.

(OK. According to the reviewers’ recommendations, the conclusion has been summarized and revised, i.e., lines from 528 to 549, which were marked in blue.)

Please accept our sincere thankfulness for your utmost effort and full support. Thank you for reviewing our manuscript.

Reviewer 2 Report

The paper needs review of English. Many parts are not clear and used terms are strange (this may be caused by too rigirous translation to English). Especially the Introduction was in some places too unclear to the Reviewer.

What do you mean by "secondary development of ABAQUS"? As I unredstand you just implemented some user-level routines. The "120 d" mean "120 days"?

The reviewer is not sure if the structure can have "diseases".

What do you mean by "vibration acceleration"?

I am not sure if the "interlaminar stress" is what you mean to study.

In any case some illustrations to the Introduction might help readers to better understand the problem you aim to study.

What is "Realization of concrete shrinkage"? Do you mean "model" or

What is "Establishment of ... model"?

In the Conclusions you declare that you made an analytical model. To what part of your work you refer?

I am not sure if "Under shrinkage" is the correct term.

Also, you verify the numerical model by comparison with an analytical model with just vague explanation that such model should be correct for your problem. Can you support this explanation by references to more sets of experimental data? You can add description of you own experiments (if any) or al least add references to other's experiemtnal results.

Author Response

Dear the reviewers,

Thank you very much for your careful and constructive comments on the manuscript (No. materials-1612399, title: Study on interlaminar stress of CRTS II slab ballastless track under concrete shrinkage and creep). Your comments are all valuable and very helpful for revising and improving our paper. According to your comments, we revised the manuscript and made correction which we hope meet with approval, and the modified parts were all marked in blue.

I hope you are satisfied with the revised version, however, if there is any question, we are willing to revise it again.

Yours sincerely,

Zhihui Zheng

Next section is the reply to reviewers’ comments:

Reviewer #2:

  1. The paper needs review of English. Many parts are not clear and used terms are strange (this may be caused by too rigirous translation to English). Especially the Introduction was in some places too unclear to the Reviewer.

(OK. According to the reviewers’ recommendations, the authors revised the grammar and sentences of this paper, and the corresponding parts were marked in blue.)

  1. What do you mean by "secondary development of ABAQUS"? As I understand you just implemented some user-level routines.

(OK. ABAQUS does not include a module to calculate shrinkage and creep of concrete. In this paper, the USDFLD and UNEXPAN subroutines are used to compile the creep and shrinkage computation modules by Fortran. Then, the user subroutine is called in the INP file of model to realize the concrete shrinkage creep calculation. According to the reviewers’ recommendations, the corresponding explanation was added, i.e., lines from 148 to 152, which was marked in blue.)

  1. The "120 d" mean "120 days"?

(Yes. According to the reviewers’ recommendations, the abbreviation of "day" has been added, i.e., line 169, which was marked in blue.)

  1. The reviewer is not sure if the structure can have "diseases".

(OK. The conclusions indicate that the ballastless track have no "diseases" under the shrinkage and creep of the ballastless track. The sliding layer cannot bear the vertical tensile force. Although a gap of the sliding layer occurs, it cannot be called a "disease". When there is a void in the sliding layer, the base plate and the girder surface will have a mutual flapping effect under the live load, which will adversely affect their service.)

  1. What do you mean by "vibration acceleration"?

(OK. According to the reviewers’ recommendations, this sentence has been revised, i.e., lines from 40 to 41, which was marked in blue.)

  1. I am not sure if the "interlaminar stress" is what you mean to study.

(OK. In this paper, the "interlaminar stress" includes the stress state of the sliding layer, interfacial stress between CA mortar and the track slab and the additional force of the shear cogging. According to the reviewers’ recommendations, some explanations were added, i.e., lines from 97 to 100, which were marked in blue.)

  1. In any case some illustrations to the Introduction might help readers to better understand the problem you aim to study.

(OK. According to the reviewers’ recommendations, some illustrations were added, i.e., Figures 1 and 2, which were marked in blue.)

  1. What is "Realization of concrete shrinkage"? Do you mean "model" or

(OK. According to the reviewers’ recommendations, "Realization of concrete shrinkage" was modified to "Calculation of concrete shrinkage and creep in ABAQUS", i.e., line 104, which were marked in blue. This section describes how to implement the shrinkage creep calculation in ABAQUS. It is not a "model", but a calculation subroutine.)

  1. What is "Establishment of ... model"?

(OK. This section is an introduction to information on the finite element model. According to the reviewers’ recommendations, "Establishment of ... model" was modified to "Finite element modeling", i.e., line 274, which was marked in blue.)

  1. In the Conclusions you declare that you made an analytical model. To what part of your work you refer?

(OK. An analytical model refers to the finite element model. "an analytical model" was modified to "a finite model", i.e., line 529, which was marked in blue. An analytical model refers to Section 3.)

  1. I am not sure if "Under shrinkage" is the correct term.

(OK. According to the reviewers’ recommendations, the title of the paper has been changed to "Numerical investigation of interlaminar stress of CRTS II slab ballastless track induced by creep and shrinkage of concrete". In addition, "Under shrinkage" was modified to "Under the effect of shrinkage", i.e., lines 22, 99 and 531, which were marked in blue.)

  1. Also, you verify the numerical model by comparison with an analytical model with just vague explanation that such model should be correct for your problem. Can you support this explanation by references to more sets of experimental data? You can add description of you own experiments (if any) or atleast add references to other's experimentalresults.

(Ok. Your recommendation is very pertinent. In this paper, a finite element model is used to study the interlaminar stresses cauesed by the shrinkage and creep of materials. The aim is to explore the potential damage that may be caused by the shrinkage and creep of ballastless track. At present, it is difficult to test the interfacial stresses. Additionally, there are no reports of experimental data or the result of a finite element model in the existing studies. Therefore, this paper only contains the finite element value. In addition, the section 4.2 was added to verify the initial stress of the structure, i.e., lines from 375 to 395, which were marked in blue. Further, the author will carry out relevant experiments in next study. )

Please accept our sincere thankfulness for your utmost effort and full support. Thank you for reviewing our manuscript.

Round 2

Reviewer 1 Report

Accept

Reviewer 2 Report

The Reviewer thinks that the paper was revised in all requested parts and can be accepted. In my opinion a final revision of English may be beneficial.

The main weak point of the paper (unavailability of other results to verify the solution) is unfortunate but the authors explained that this is not possible just now. The Reviewer understands their explanation.